# Tricine as a Novel Cryoprotectant with Osmotic Regulation, Ice Recrystallization Inhibition and Antioxidant Properties for Cryopreservation of Red Blood Cells

**DOI:** 10.3390/ijms23158462

**Published:** 2022-07-30

**Authors:** Xiangjian Liu, Yuying Hu, Wenqian Zhang, Deyi Yang, Yuxin Pan, Marlene Davis Ekpo, Jingxian Xie, Rui Zhao, Songwen Tan

**Affiliations:** Xiangya School of Pharmaceutical Sciences, Central South University, Changsha 410013, China; liuxiangjiancsu@163.com (X.L.); nx_yuyinghu@126.com (Y.H.); 8305200509@csu.edu.cn (W.Z.); yangdeyi1999@163.com (D.Y.); zoeypan0215@163.com (Y.P.); 217208001@csu.edu.cn (M.D.E.); xiaoxian.xie@csu.edu.cn (J.X.); zr2498176546@163.com (R.Z.)

**Keywords:** transfusion therapy, cryopreservation, tricine, red blood cells, cryoprotectant, mathematical model

## Abstract

The cryopreservation of red blood cells (RBCs) plays a key role in blood transfusion therapy. Traditional cryoprotectants (CPAs) are mostly organic solvents and may cause side effects to RBCs, such as hemolysis and membrane damage. Therefore, it is necessary to find CPAs with a better performance and lower toxicity. Herein, we report for the first time that N-[Tri(hydroxymethyl)methyl]glycine (tricine) showed a great potential in the cryopreservation of sheep RBCs. The addition of tricine significantly increased the thawed RBCs’ recovery from 19.5 ± 1.8% to 81.2 ± 8.5%. The properties of thawed RBCs were also maintained normally. Through mathematical modeling analysis, tricine showed a great efficiency in cryopreservation. We found that tricine had a good osmotic regulation capacity, which could mitigate the dehydration of RBCs during cryopreservation. In addition, tricine inhibited ice recrystallization, thereby decreasing the mechanical damage from ice. Tricine could also reduce oxidative damage during freezing and thawing by scavenging reactive oxygen species (ROS) and maintaining the activities of endogenous antioxidant enzymes. This work is expected to open up a new path for the study of novel CPAs and promote the development of cryopreservation of RBCs.

## 1. Introduction

Transfusion of red blood cells (RBCs) is an effective and widely used method for traumas, hematological malignancies, leukemia and so on. It is reported that ~24 million blood units are used for transfusion every year in the USA [1]. The huge demand for RBCs requires excellent storage methods. In general, RBCs can only be stored at 4 °C for up to 42 days with the conventional method. After 21 days, RBCs are considered to be aged and may suffer from hemolysis, the disruption of membrane stability and the loss of other functions [2,3]. The transfusion of aged RBCs may cause lung [4] and liver [5] injury, and even increase the risk of death [6]. Therefore, it is necessary to improve the storage method of RBCs. Cryopreservation provides a novel perspective for the long-term storage of cells. Under ultra-low temperatures (−80 °C or −196 °C), the physiological activity of the cells will be significantly reduced or even stop [7]. As a result, cells can maintain their viability for even decades through cryopreservation [8].

However, there are some challenges in cryopreservation. In 1972, Mazur et al. [9] proposed that cells mainly suffered from osmotic damage and mechanical damage during cryopreservation, and this hypothesis is still prevalent nowadays [10]. Osmotic damage is caused by extracellular ice formation that can induce the severe dehydration of cells. Mechanical damage is related to the large intracellular ice formed by recrystallization during thawing, and it may destroy the membrane severely [11]. Recently, oxidative damage has also attracted many researchers’ attention [12]. Cells may produce more reactive oxygen species (ROS) during cryopreservation, which can cause DNA damage, protein oxidation and lipid peroxidation [13]. Therefore, cryoprotectants (CPAs) must be added to samples to reduce the damage in cryopreservation. Glycerol is the only CPA that has been approved for the clinical cryopreservation of RBCs [14]. However, to obtain satisfactory cell recovery, very high concentrations of glycerol must be added (20–40% *w*/*v*) [15]. Such high concentrations of glycerol induce the hemolysis of RBCs [16], and the procedure of deglycerolization after cryopreservation can be very complex [8]. Hydroxyethyl starch (HES) is also commonly used in the cryopreservation of RBCs, which could stabilize the cell membrane and inhibit the formation of ice [17]. Compared to glycerol, HES is safer and does not affect the properties of RBCs [18]. However, HES often shows a poor efficiency in the cryopreservation of RBCs. Even with very high concentrations of HES, the thawed RBCs recovery is still less than 40% [7,19].

Therefore, it is necessary to search for biocompatible, nontoxic and effective CPAs. Trehalose [16], polyampholyte [20], betaine [8], sodium hyaluronate [21] and block copolymer worms [7], etc. have been reported to be suitable for the cryopreservation of RBCs. However, these novel CPAs are not effective enough when used alone, possibly because they generally only mitigate one or two types of cryoinjuries. To reach a satisfactory RBC recovery, they often need to be combined with other CPAs in cryopreservation. For example, when trehalose was used alone in cryopreservation, the cell recovery was only ~30%. Only when trehalose was combined with proline did the cell recovery reach ~90% [22]. The combined use of CPAs increases cost and operational complexity.

N-[Tri(hydroxymethyl)methyl]glycine (Tricine) is a glycine derivative (Figure 1), and it has been widely used in biomedicine and physiology [23]. For example, Tricine-SDS-PAGE is considered as the preferred electrophoretic system for the resolution of proteins smaller than 30 kDa [24]. In the cryopreservation of boar sperm, tricine has been used as a buffer in diluents, and sperm motility could be maintained [25]. However, tricine has not been used for cryopreservation specifically, nor has the mechanism of tricine in cryopreservation been studied. In previous studies, glycine, the parent compound of tricine, has been proven to be efficient in inhibiting ice formation and protecting cells from osmotic damage [10]. Additionally, tricine could remove ROS effectively, and the rate constant of hydroxyl radicals with tricine is up to 1.6 × 10^9^ M^−1^·S^−1^ [26]. Inspired by these studies, it can be expected that tricine may show a great performance in cryopreservation. 

This study demonstrated for the first time that tricine could simultaneously reduce osmotic, mechanical and oxidative damages during cryopreservation. A satisfactory sheep RBC recovery was achieved after cryopreservation without the addition of organic solvents. The results of morphology, erythrocyte sedimentation rate (ESR), ATPase activity and content of hemoglobin (Hb) indicated that the properties of the thawed RBCs were appreciably preserved. Mathematical modeling analysis showed the advantages of tricine over other CPAs. These findings indicated the great potential of tricine in the cryopreservation of RBCs and may be beneficial for clinical transfusion.

## 2. Results

### 2.1. The Feasibility of Tricine in Cryopreservation

The RBCs were incubated in different solutions at 4 °C for 24 h, and the RBCs’ recovery was considered as an indicator of biocompatibility [24]. As shown in Figure 2A, except for the 8% tricine group (93.6 ± 0.7%), the RBCs’ recovery in other concentrations of tricine solutions was more than 95%. However, the RBCs’ recovery in the glycerol group was only 74.9 ± 2.6%, significantly lower than that in any other group (*p* < 0.001). The result is consistent with the conclusion that high concentrations of glycerol easily cause hemolysis [15]. HES is often used as a plasma substitute, indicating its good safety for blood [27]. Moreover, the RBCs’ recovery was 96.0 ± 1.0% in 13% HES, showing a great biocompatibility.

To determine the efficiency of tricine in cryopreservation, RBCs were frozen and thawed in different CPA solutions. As shown in Figure 2B, the thawed RBCs’ recovery was positively correlated with the concentration of tricine within a certain range. The optimal concentration of tricine was 6% and the thawed RBCs’ recovery was up to 81.2 ± 8.5% in this group. Glycerol showed the best efficiency, and the thawed RBCs’ recovery was 89.5 ± 1.7%. HES performed poorly with a thawed RBCs’ recovery of 39.7 ± 5.3%, showing its low efficiency when used alone for cryopreservation.

### 2.2. The Ability of Osmotic Regulation

To test the osmotic regulation ability of tricine, a 2.9% NaCl solution was made to mimic the hypertonic environment in cryopreservation. After incubating in 2.9% NaCl solution for 72 h, the RBCs’ recovery was only 87.5 ± 3.3%, indicating that many RBCs died due to osmotic damage. In contrast, the addition of 2% tricine could significantly increase the RBCs’ recovery to 93.5 ± 1.0% (Figure 3A,B), showing it was beneficial for RBCs to survive in a hypertonic environment. The RBCs’ morphology in 2.9% NaCl became thorny, indicating that the remaining RBCs were undergoing serious dehydration and deformation. The addition of 2% tricine could make RBCs maintain the normal spherical shape (Figure 3C). In previous studies, glycine was considered to be a natural osmoprotectant [10,28,29]. In hypertonic environments, glycine can permeate the cell membrane and become highly accumulated in cells [30,31]. Thus, the outflow of intracellular water and the dehydration of cells could be mitigated. As a derivative of glycine, tricine was considered to maintain an osmotic balance through a similar mechanism.

### 2.3. Reducing the Mechanical Damage from Ice

Differential scanning calorimetry (DSC) was used to analyze the bound water during freezing and thawing (Figure 4A). The endothermic peak area of pure water was larger than that of the tricine solutions, indicating that bound water arose in tricine solutions. The ratio of bound water was proportional to the concentrations of tricine (Figure 4B), and it was up to 53.6% in the 6% tricine solution.

To further study the ice recrystallization inhibition (IRI) activity of tricine, the size of ice was obtained under a polarizing microscope by splat assay [7]. The ice crystals in the PBS group (blank control) were the largest (Figure 4C), which would cause serious mechanical damage to cells in cryopreservation. In contrast, the size of ice in each tricine group was obviously smaller and negatively correlated with the concentration of tricine (Figure 4D–G). The mean largest grain size (MLGS) was used as an index for quantitative analysis of the IRI activity (Figure 4H). The MLGS of the PBS group was up to 107.55 ± 4.14 μm, while after the addition of a very low concentration of tricine (1%), the MLGS could be significantly reduced to 98.30 ± 3.11 μm (*p* < 0.05). With the increase in concentrations of tricine, the MLGS became much smaller. In 6% tricine, the MLGS was only 53.29 ± 0.52 μm, approximately 50% lower than those in the PBS group. The whole processes of ice in PBS and 6% tricine were shown in Appendix A, indicating that ice recrystallization could be greatly inhibited by tricine.

### 2.4. Resistance to Oxidative Damage

The α, α-diphenyl-β-picrylhydrazyl (DPPH) assay is commonly used to test antioxidative capacity [32]. In general, the antioxidative capacity is represented as an equivalent concentration of Trolox, which is an antioxidant with a specific capacity [33]. The standard curve of DPPH clearance and Trolox concentrations showed a good linearity (Appendix A). The antioxidative capacity of HES was up to 11.16 ± 1.27 μgTrolox/mL, while that of PBS was not detectable. The antioxidant capacities of glycerol and tricine were 0.80 ± 0.29 μgTrolox/mL and 4.35 ± 0.26 μgTrolox/mL, respectively, which were lower than that of HES (Figure 5A).

To test the endogenous antioxidant properties of post-thawed RBCs, the activities of catalase and superoxide dismutase (SOD) were measured. For catalase activity, the HES group (42.46 ± 5.72 U/gHb) was significantly higher than the PBS group (28.05 ± 0.24 U/gHb) and glycerol group (29.78 ± 1.35 U/gHb). In addition, the tricine group (33.37 ± 3.42 U/gHb) and control group (32.87 ± 5.00 U/gHb) were similar (Figure 5B). For SOD activity, the glycerol group (7.94 ± 0.08 × 10^4^ U/gHb), tricine group (7.49 ± 0.16 × 10^4^ U/gHb) and HES group (8.33 ± 0.47 × 10^4^ U/gHb) were all obviously higher than the PBS group (6.81 ± 0.35 × 10^4^ U/gHb) and control group (6.12 ± 0.35 × 10^4^ U/gHb) (Figure 5C).

RBCs’ membranes undergo lipid peroxidation (LPO) by ROS. Malondialdehyde (MDA) is a marker product of LPO, and the content of MDA can be used as an indicator of oxidative damage [34]. After cryopreservation, the content of MDA in the PBS group was up to 15.30 ± 1.13 nmol/gHb, significantly higher than that in the glycerol group (2.48 ± 0.92 nmol/gHb), tricine group (2.30 ± 0.85 nmol/gHb) and HES group (0.99 ± 0.30 nmol/gHb). The content of MDA was not detectable in the control group.

### 2.5. The Properties of Thawed RBCs

The morphology, Erythrocyte sedimentation rate (ESR), ATPase activities and content of hemoglobin (Hb) were used to test the properties of the thawed RBCs. After freezing and thawing, the RBCs in the PBS group were obviously dehydrated and shrunk (Figure 6A), indicating that the RBCs have suffered from severe osmotic damage. In contrast, the RBCs in the tricine group maintained a normal spherical shape without major deformations (Figure 6B). As shown in Figure 6C, the ESR of the tricine group was slightly higher than that of the control group within 1 h, 4 h, 7 h and 10 h, but there was no significant difference between them. The activities of ATPases were used as indexes to determine the RBCs’ biochemical properties. After cryopreservation with tricine, the activities of Na^+^/K^+^ ATPase and Ca^2+^/Mg^2+^ ATPase were similar to those in the control group (Figure 6D), indicating that the thawed RBCs could transport ions normally and maintain physiological function. The content of Hb indicated the oxygen-carrying capacity, and there was no significant difference between the tricine group and control group (Figure 6E).

### 2.6. Comparing CPAs by Mathematical Models

To show the advantage of tricine in cryopreservation, the technique for order preference by similarity to an ideal solution (TOPSIS) was used to quantitatively compare the efficiency of HES, glycerol and tricine. TOPSIS is a classic mathematical model for multiple-criteria decision making, which can make full use of attribute information and provide a ranking of alternatives [35,36]. Three related criteria of CPAs, including the thawed RBCs’ recovery, concentration and biocompatibility, were used to form an initial decision matrix, as shown in Table 1. The thawed RBCs’ recovery and biocompatibility were considered as the benefit attributes while the concentration was the cost attribute. The results of the TOPSIS model, including the closeness coefficient and rank of the CPAs, are shown in Table 2. Tricine ranked first with a closeness coefficient of 0.853, while glycerol and HES were 0.622 and 0.355, respectively.

To further study the preference of CPAs in different aspects, the analytic hierarchy process (AHP) [37] and rank-sum ratio (RSR) [38] models were used for analysis. AHP is one of the most widely used multiple criteria decision-making tools and has been used successfully in many fields [39]. The objective, criteria and plans in AHP are shown in Appendix A, and the results are shown in Appendix A. The weights of the three CPAs (glycerol, HES and tricine) were close (Appendix A), while tricine still showed the best performance. RSR is a comprehensive evaluation method and it can divide alternatives into different levels [38]. The origin data of CPAs and indexes of the RSR model are shown in Appendix A, and the CPAs were divided into three levels through Probit. Tricine was considered as “excellent” while other CPAs were considered as good (Appendix A).

## 3. Discussion

Although cryopreservation is an effective way to prolong the storage time of RBCs, there are still some limitations to be addressed. During freezing and thawing, extracellular ice formation will increase the osmotic pressure and make cells dehydrated; this is known as osmotic damage. We report the great ability of tricine to regulate osmotic balance, indicating its efficiency in reducing osmotic damage in cryopreservation. The mechanism of tricine in osmotic regulation might be similar to that of glycine: in a hypertonic environment, tricine might enter RBCs to maintain a high intracellular concentration. As a result, the dehydration and osmotic shock are mitigated.

The mechanical damage from ice also severely affects RBCs. During thawing, the ice can grow larger through recrystallization, which can be lethal to cells [40]. Ice recrystallization relies on the aggregation of water molecules and the development of a hydrogen bond network (HBN) [8]. Some amino acids, such as proline and glycine, have IRI activity to reduce the mechanical damage in cryopreservation [10]. These amino acids are hydrophilic and have water-binding capacity. Thus, the HBN among water molecules is destroyed and the ice growth is inhibited. It is well-known that tricine has three hydroxyl groups and one carboxyl group, and this special structure gives it a stronger hydrophilicity to form bound water. In our study, the water-binding capacity of tricine was higher than that of proline and glycine, indicating its better efficiency in inhibiting ice growth.

Additionally, high levels of ROS are produced and cause oxidative damage to RBCs during cryopreservation. On the one hand, we have found tricine to have a good antioxidative capacity. On the other hand, the activities of endogenous antioxidative enzymes could be maintained by tricine. Catalase and SOD have been proven to be the important enzymes to scavenge H_2_O_2_ and superoxide anions, respectively [41,42]. We showed that catalase is damaged during freezing and thawing, and that the addition of tricine could maintain its property. For SOD, the activity increased after cryopreservation in tricine. In previous studies, the addition of antioxidants, such as ascorbic acid and malate dehydrogenase, could increase SOD activity after cryopreservation [43,44]. The results further demonstrated that tricine has a similar ability. The MDA content was used to directly demonstrate the oxidative damage in cryopreservation. The results indicated that RBCs’ membranes are severely destroyed by the ROS produced in cryopreservation. Tricine could not only directly remove ROS, but also maintain the activities of catalase and SOD. Therefore, the oxidative damage to RBCs was mitigated. It must be noted that classic CPAs, such as glycerol and HES, have a similar ability to reduce antioxidant damage.

Based on the above discussion, the mechanism of tricine in the cryopreservation of RBCs is shown in Figure 7.

The properties of thawed RBCs are essential for the success of transfusion [45]. In this study, the morphology, ESR and ATPase activities and the content of Hb were considered as indexes for properties. First, the normal morphology of RBCs is the basis for maintaining their function. Second, ESR is a factor for hemorheology. The normal ESR fluctuates within a narrow range, and an abnormally raised ESR indicates impaired rheological properties of RBCs. Third, the Na^+^/K^+^ ATPase and Ca^2+^/Mg^2+^ ATPase in RBCs’ membranes are important to maintain the intracellular gradients of ions. Fourth, the content of MDA indicates the oxygen-carrying capacity. The results showed all the indexes of RBCs cryopreserved in tricine were maintained normally, and these RBCs might be used successfully in clinic transfusion.

To further compare the performance of tricine and other classic CPAs quantitatively, we have used several mathematical models for analysis. The TOPSIS, AHP and RSR models have proved the great efficiency of tricine in cryopreservation in different aspects. It must be noted that the results of the mathematical models only mean that the overall performance of tricine was better according to the selected criteria, and the results do not mean that tricine could replace HES or glycerol in cryopreservation. The choice of CPA should be based on the study requirements.

In summary, tricine could significantly improve the thawed RBCs’ recovery without the addition of organic solvents. The RBCs’ morphology, ESR, ATPase activities and the content of Hb were normal after cryopreservation. During cryopreservation, tricine could maintain osmotic balance, increase the ratio of bound water, inhibit ice recrystallization and counter oxidative damage. Mathematical modeling analysis demonstrated the good performance of tricine in the cryopreservation of RBCs. Therefore, tricine is expected to open up a new way for cryopreservation. It must be noted that the RBCs used in this study were from sheep, and more experiments are needed to prove the effectiveness of tricine on human RBCs.

## 4. Materials and Methods

### 4.1. RBCs Preparation and Recovery Test

Sheep RBCs (Hongquan, Guangzhou, China) were collected in centrifuge tubes with 0.01M PBS and shaken to mix evenly. Then, RBCs were centrifuged at 1980× *g* for 3 min, and the supernatant that contained anticoagulant, white blood cells and plasma was removed. The above operation was repeated twice to obtain washed RBCs. To measure RBC recovery, the samples that contained RBCs were centrifuged at 1980× *g* for 3 min, and the absorbance of the supernatant was measured at 450 nm (SHIMADZU, UV2600, Kyoto, Japan). The RBC recovery can be calculated by the following equation [10]:(1)$$\mathrm{Hemolysis}\%=\frac{A-A_{0}}{A_{1}-A_{0}}\times 100\%$$
(2)RBCs recovery %=1−Hemolysis(%)
where A is the absorbance of the measured sample and A_0_ and A_1_ are the absorbance of equal amounts of RBCs in PBS and deionized water, respectively.

### 4.2. Biocompatibility Test

The CPA solutions, including 20% glycerol (Sinopharm, Shanghai, China), 13% HES (Macklin, Shanghai, China) and 2%, 4%, 6%, 8% tricine (Yuanye Bio-Technology, Shanghai, China) solutions were all formulated in PBS. Equal amounts of washed RBCs (100 μL) were added into PBS and the CPA solutions as the control group and experimental groups, respectively. The samples were stored at 4 °C for 48 h and resuspended every 24 h. The RBCs recovery was considered as an indicator of biocompatibility [16], and it could be obtained by the method described in Section 4.1.

### 4.3. SEM Analysis

The RBCs were fixed in 2.5% glutaraldehyde for 12 h. Then, the RBCs were washed with PBS three times to remove the residual glutaraldehyde and were fixed again in 1% osmic acid for 1 h. Subsequently, the osmic acid was removed and RBCs were washed with PBS three times. The RBCs were dehydrated in 50%, 70%, 90% and 100% ethanol for 15 min, respectively. After that, the RBCs were incubated in 50%, 70%, 90% and 100% isoamyl acetate for 15 min. After supercritical drying, the RBCs were gold-coated and photographed under scanning electron microscopy (Hitachi, S-3400N, Tokyo, Japan).

### 4.4. Cryopreservation of RBCs

All the solutions were prepared in PBS at the desired concentration. Equal amounts of washed RBCs (100 μL) were mixed with 5 mL of the prepared solution or PBS as the experimental group and the control group, respectively. After incubation at room temperature for 30 min, the samples were immersed directly into liquid nitrogen until completely frozen for at least 20 min [46]. The frozen RBCs were rewarmed in a 37 °C water bath. The thawed RBCs’ recovery can be obtained by the method in 4.1.

### 4.5. Osmotic Regulation Test

Equal washed RBCs (100 μL) were exposed to 5 mL solutions containing 2.9% NaCl and 2.9% NaCl with 2% tricine, respectively. Then, the samples were stored at 4 °C and the RBCs were resuspended every 24 h. After 72 h, the RBCs’ recovery could be calculated by the absorbance of the supernatant and the morphology could be observed by SEM as described above [10].

### 4.6. Splat Assay

The samples of PBS and different concentrations of tricine (1%, 2%, 4% and 6%, respectively) were prepared as described above. The splat assay was adapted from a previous study [7]. Briefly, a 6 μL sample was dropped from a height of ~1.4 m onto a pre-cooled glass slide. Then, the glass slide was moved onto a cold stage (Huitong, LTM-190H, Shanghai, China) that was maintained at −8 °C. The sample was annealed at this temperature for 20 min. The ice growth was observed under a polarizing microscope (Huitong, XPF-550, Shanghai, China), and images were taken every 1 min. The MLGS was obtained by measuring the mean size of the 10 largest ice crystals in the field of view.

### 4.7. DSC Test

DSC was used to investigate the water binding capacity of tricine [10]. Briefly, accurate weight samples (~10 mg) were placed in crucibles. Then, they were sealed and transferred to the calorimeter sample chamber (Beijing Henven, HSC-4, China). The samples were cooled from room temperature to −35 °C at −3 °C /min, held for 5 min, and then rewarmed to 10 °C at 1 °C /min. The heat flow (w/g) was monitored. The total water content (w_tc_), the frozen water content (w_f_) and the bound water content (w_b_) could be calculated by the following equations [8]:
(3)$$w_{\mathrm{tc}}\;=\;m_{w}/m$$
(4)$$w_{f}\;=\;\Delta H/\Delta H_{w}$$
(5)$$w_{b}\;=\;w_{t}\;-\;w_{f}$$
where m_w_ and m represent the water mass and the total mass of each sample, respectively, and ΔH and ΔH_w_ are the melting enthalpies of each sample and pure water, respectively, determined by integration from the onset temperature to the end temperature of the heat flow.

### 4.8. Antioxidant Assay

The DPPH free radical scavenging method (Nanjing Jiancheng, A153-1-1, Nanjing, China) was used to estimate the antioxidant activity of tricine. The antioxidant enzyme activities of the RBCs, including superoxide dismutase (SOD) and catalase, were measured by the hydroxylamine method (Nanjing Jiancheng, A001-1-1, China) and ultraviolet method (Nanjing Jiancheng, A007-2-1, China). The lipid peroxidation of thawed RBCs after cryopreservation was measured by the malondialdehyde level (Nanjing Jiancheng, A003-1-1, China). All the measurement methods are described in the manufacturers’ instructions in detail.

### 4.9. ESR Analysis

The Westergren method was used to determine the ESR of the RBCs [47]. A 50 μL aliquot of RBCs washed after cryopreservation was added to 2 mL PBS. The sample was transferred to a Westergren tube and the level of the sample was controlled to the 0-scale mark. A 50 μL aliquot of fresh RBCs was treated in the same way as the control group. The ESR of the samples was obtained at 1 h, 4 h, 7 h and 10h.

### 4.10. ATPase Activities

The Na^+^/K^+^-ATPase, Ca^2+^/Mg^2+^-ATPase activities assays were performed with fresh RBCs and thawed RBCs after cryopreservation. The ATPases activities were measured using an inorganic phosphorus method (Nanjing Jiancheng, A070-5-4, China) and the operation was carried out following the kit’s instructions.

### 4.11. Content of Hb

The concentration of Hb (mgHb/mL) was tested by the HICN colorimetric method (Nanjing Jiancheng, C021-1-1, China). The number of RBCs per milliliter was determined by hemocytometer. Thus, the content of Hb in RBCs (mgHb/10^9^ RBCs) could be obtained.

### 4.12. Mathematical Model Analysis

The TOPSIS model was used to evaluate the performance of tricine, glycerol and HES in cryopreservation [35,48]. An evaluation matrix **X** consisting of m alternatives and n criteria was created by Equation (6). Then, each element $x_{ij}$ was normalized into a corresponding element $S_{ij}$ in the matrix **S** by the Equations (7)–(9). The negative ideal solutions (NIS) $S_{j}^{-}$ and positive ideal solutions (PIS) $S_{j}^{+}$ were calculated by Equations (10) and (11), respectively. The Euclidean distance between the alternative and PIS (D^+^)/NIS (D^−^) was calculated based on Equations (12) and (13). Last, the relative closeness of the *i*th alternative C*_i_* with respect to the ideal solutions was obtained as Equation (14), and it is proportional to the performance of the CPAs.
(6)$$X={\left(x_{ij}\right)}_{m\times n}=\left(\begin{array}{cccc} x_{{}^{11}} & x_{12} & \cdots & x_{1n}\\ x_{21} & x_{22} & \cdots & x_{2n}\\ \vdots & \vdots & \vdots & \vdots \\ x_{m1} & x_{m2} & \cdots & x_{mn} \end{array}\right)$$
(7)$$S={\left(s_{ij}\right)}_{m\times n}=\left(\begin{array}{cccc} s_{{}^{11}} & s_{12} & \cdots & s_{1n}\\ s_{21} & s_{22} & \cdots & s_{2n}\\ \vdots & \vdots & \vdots & \vdots \\ s_{m1} & s_{m2} & \cdots & s_{mn} \end{array}\right)$$
where
(8)$$S_{ij}=\frac{x_{ij}}{\sqrt{\sum_{i=1}^{m}{\left(x_{ij}\right)}^{2}}},\;\mathrm{for}\;\mathrm{benefit}\;\mathrm{attribute}\;x_{\mathrm{ij}},$$
and
(9)$$S_{ij}=\frac{x_{j}^{\max}-x_{ij}}{\sqrt{\sum_{i=1}^{m}{\left(x_{j}^{\max}-x_{ij}\right)}^{2}}},\;\mathrm{for}\;\cos t\;\mathrm{attribute}\;x_{\mathrm{ij}},\;\mathrm{where}\;x_{j}^{\max}\;=\;\overset{m}{\underset{i}{\max}}\;x_{ij}$$
(10)$$S_{j}^{-}=\overset{m}{\underset{i=1}{\min}}\;S_{ij}$$
(11)$$S_{j}^{+}=\overset{m}{\underset{i=1}{\max}}\;S_{ij}$$
(12)$$D_{i}^{+}=\sqrt{\sum_{j=1}^{n}{\left(S_{j}^{+}-s_{ij}\right)}^{2}}$$
(13)$$D_{i}^{-}=\sqrt{\sum_{j=1}^{n}{\left(S_{j}^{-}-s_{ij}\right)}^{2}}$$
(14)$$C_{i}=\frac{D_{i}^{-}}{D_{i}^{-}+D_{i}^{+}}$$

### 4.13. Data Analysis

All statistics and calculations were determined by Origin 2022. The results of the RBC experiments are represented by the mean ± standard deviation of three independent experiments. Statistical significance determination used the student’s *t*-test. A *p*-value of less than 0.05 was considered statistically significant. * indicates *p* < 0.05, ** indicates *p* < 0.01, *** indicates *p* < 0.001.

## Figures and Tables

**Figure 1 ijms-23-08462-f001:**
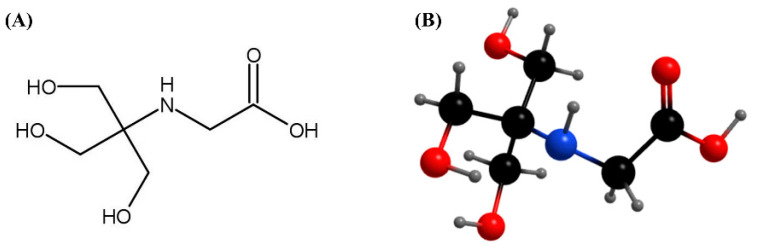
The 2D (**A**) and 3D (**B**) molecular structure of tricine.

**Figure 2 ijms-23-08462-f002:**
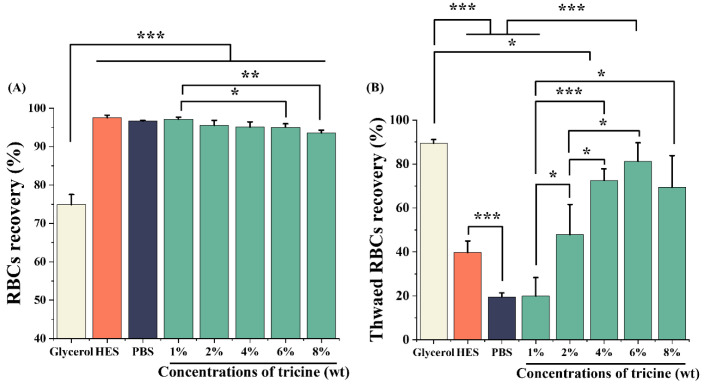
The biocompatibility (**A**) and thawed RBCs’ recovery (**B**). The concentrations of glycerol and HES were 20% and 13%, respectively. * *p* < 0.05; ** *p* < 0.01; *** *p* < 0.001.

**Figure 3 ijms-23-08462-f003:**
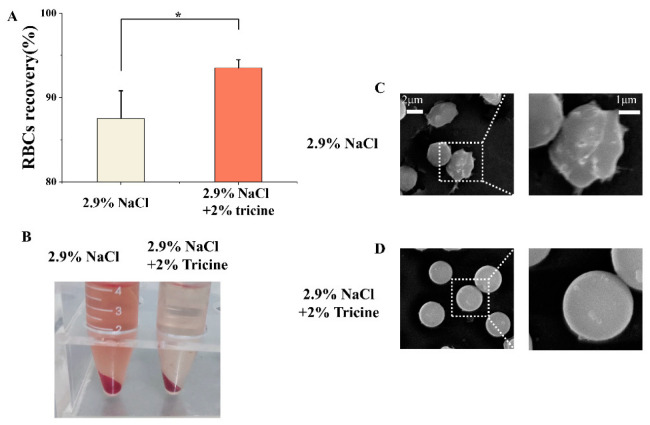
The osmotic regulation of tricine in a hypertonic environment. The recovery (**A**) and representative images (**B**) of RBCs incubated in 2.9% NaCl and 2.9% NaCl + 2% tricine solutions for 72 h. The representative SEM images of RBCs’ morphology after incubation in 2.9% NaCl (**C**) and 2.9% NaCl + 2% tricine (**D**) solutions for 72 h. Three replicates of each sample were tested and the data are shown as mean ± SD. * indicates *p* < 0.05.

**Figure 4 ijms-23-08462-f004:**
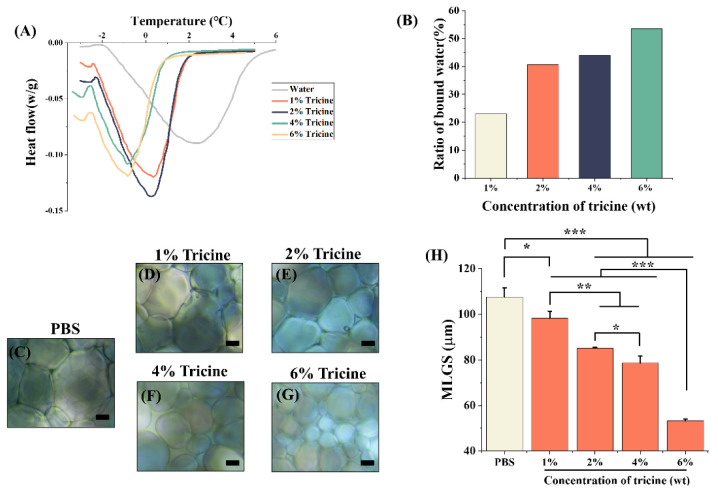
The ability to reduce the mechanical damage from ice. (**A**) The heat flow during the melting process. (**B**) The ratio of bound water in different concentrations of tricine. The representative images of ice crystals in (**C**) PBS, (**D**) 1% tricine, (**E**) 2% tricine, (**F**) 4% tricine and (**G**) 6% tricine. (**H**) Quantitative analysis of IRI activity through MLGS. MLGS = mean largest grain size. * *p* < 0.05; ** *p* < 0.01; *** *p* < 0.001. Scale bar = 20 μm.

**Figure 5 ijms-23-08462-f005:**
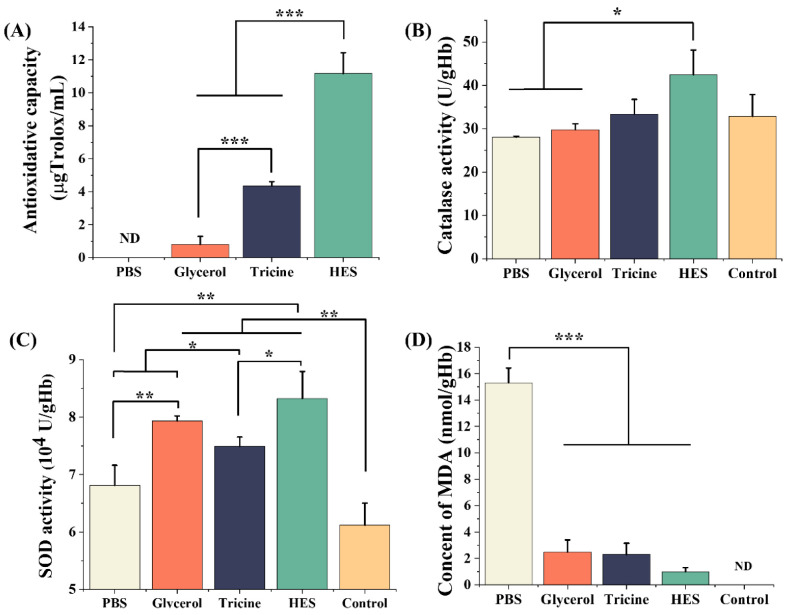
The ability to reduce oxidative damage. (**A**) The antioxidative capacity evaluated by DPPH assay. The catalase (**B**) and SOD (**C**) activities of RBCs cryopreserved in different solutions. (**D**) The lipid peroxidation of RBCs, the results are presented as the content of MDA. The control group comprised fresh RBCs without cryopreservation. The concentrations of glycerol, tricine and HES were 20%, 6% and 13%, respectively. Three replicates of each sample were tested and the data are shown as mean ± SD. * indicates *p* < 0.05, ** indicates *p* < 0.01, *** indicates *p* < 0.001.

**Figure 6 ijms-23-08462-f006:**
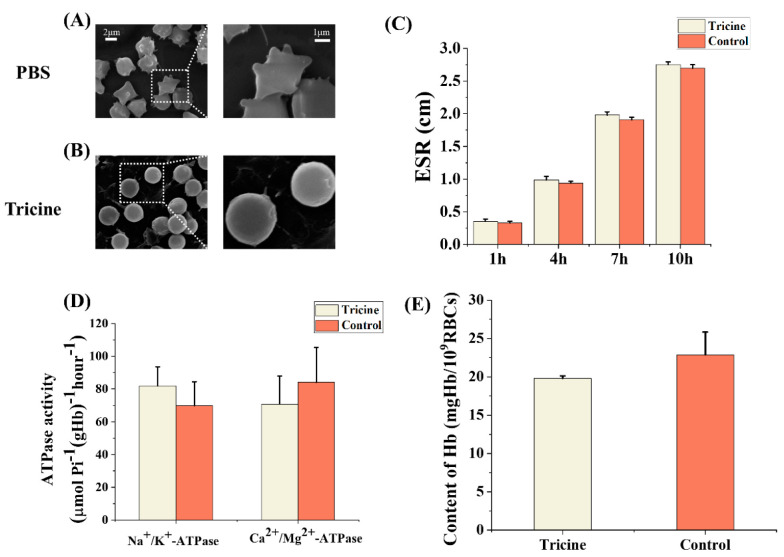
The properties of RBCs after cryopreservation. The representative thawed RBCs’ morphologies when cryopreserved in PBS (**A**) and tricine (**B**). The ESR (**C**), ATPase activities (**D**) and content of Hb (**E**) of thawed RBCs in the control group and tricine group. The concentration of tricine was 6%wt in all experiments. The control group comprised the fresh RBCs without cryopreservation. Three replicates of each sample were tested and the data are shown as mean ± SD. ESR = erythrocyte sedimentation rate. Hb = hemoglobin.

**Figure 7 ijms-23-08462-f007:**
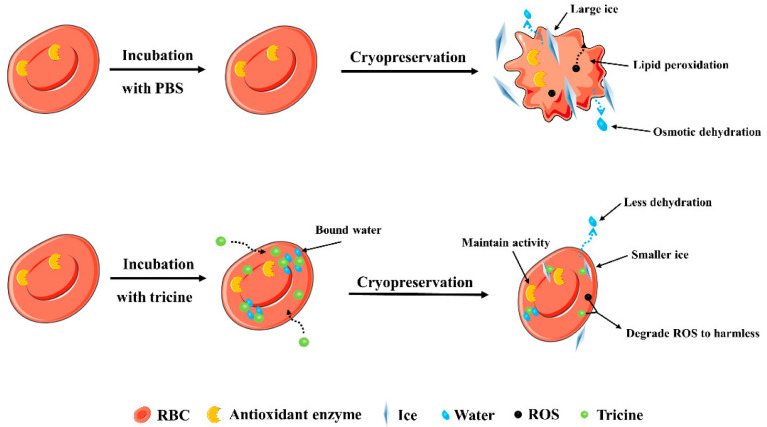
The proposed mechanism of tricine in cryopreservation.

**Table 1 ijms-23-08462-t001:** Three alternative CPAs and five criteria.

Types of CPAs	Concentration (%wt)	Thawed RBCs Recovery (%)	Biocompatibility(%)
Glycerol	20.0	89.5	74.9
HES	13.0	39.7	97.5
Tricine	6.0	81.2	95.8

**Table 2 ijms-23-08462-t002:** The result of TOPSIS.

Types of CPAs	Closeness Coefficient	Rank
Glycerol	0.622	2
HES	0.355	3
Tricine	0.853	1

## Data Availability

All experimental data within the article are available from the corresponding author upon reasonable request.

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
