# Peer review of "Tricine as a Novel Cryoprotectant with Osmotic Regulation, Ice Recrystallization Inhibition and Antioxidant Properties for Cryopreservation of Red Blood Cells"

_ijms, 2022, doi:10.3390/ijms23158462_

Round 1
Reviewer 1 Report
The paper explores the potential of tricine to act as a cryoprotectant for the cryopreservation of red blood cells.
The authors set their findings against a backdrop of the potential use of tricine as a CPA for the cryopreservation of human RBCs (hRBC). This is set out in the Introduction and Discussion sections of the manuscript. However, it is not made clear until the Methods and Materials that the current study is on ovine RBCs. This is important as the translation of CPA effectiveness from one species to another, particularly in the case of red cells, is often problematic.
· The authors need to address this earlier in the manuscript (title, abstract or introduction)
The authors choose to use DMSO and HES as comparators against which to measure the effectiveness of their putative CPA, rather than glycerol which is the only CPA licensed for clinical use for cryopreserved hRBCs. Published literature suggest that DMSO is unsuitable for hRBC cryopreservation and, while HES has been used successfully to cryopreserve hRBCs, the optimum concentration is generally far lower than that used in the present study.
· The authors need to address these issues and the question of why glycerol was omitted as a comparator
The authors use HES and DMSO to demonstrate that tricine provides comparable biocompatibility and post-thaw recovery to these two CPAs respectively. However, when assessing tricine’s ability to reduce ROS, limit peroxidation and maintain antioxidant capacity the comparator use is PBS which is not a CPA. In so doing, it makes it impossible to assess whether tricine as a CPA is better or worse than any of the alternative CPAs use for red cell cryopreservation.
· The authors need to explain and justify their choice of a non-CPA as the comparator and why, in the absence of results from alternative CPAs, they are justified in saying that the Tricine performs better than HES or DMSO.
Though I am unfamiliar with the mathematical modelling used in this paper, which appears to be translated from the social sciences, I would question some of the assumptions used in the parameters chosen for analysis. Firstly, the cost parameter appears to be based solely on the price provided by the authors chosen supplier. Costs for DMSO and HES vary widely depending on the grade used and whether or not it has regulatory clearance for human application (for example, a brief search of the internet produced a range from $0.044/gm to $1/gm). Secondly, the authors have assumed a set of optimal concentrations only one of which, for tricine, appears to be based on experimental evidence. The literature suggests optimal concentrations of HES for hRBCs lie between 11.5 and 17%, while the choice of 10% DMSO is presumably based on the generally-accepted, non-optimized concentration used for nucleated cells. It is therefore likely that changes in the assumptions based on different figures for these two parameters would alter the results of the TOPSIS analysis.
· At the very least the authors need to explain the basis for choosing the values of these two parameters and the effect that varying these would have on their analysis.
· The authors should again consider, in the light of the potential for different assumptions to alter the outcome of the modelling, whether the statement that tricine is a “better” CPA for RBC cryopreservation, is still valid.
The RSR model (based on Tables S7 and S8) show a Probit value for Tricine of 5.674 – below the threshold level (>6) for “excellent” (table S9). The authors assert in the manuscript that Tricine is ranked as excellent when in fact it appears from S8 and S9 that its ranking is on a par with the other CPAs.
· The authors need to explain this apparent discrepancy.
In summary, whilst the paper is well written and the data on biocompatibility and recovery is persuasive, the use of PBS as a comparator in the metabolic studies is problematic and the mathematical modelling is unfortunately unconvincing.
Author Response
Please see the file "Response to Reviewer 1"

Reviewer 2 Report
The manuscript by Liu et al. deals with the cryoprotectant effect of tricine for RBCs storage under ultra-low temperatures. Thus, the authors compared RBCs recovery obtained in tricine added samples with the ones obtained with other cryoprotectants (HES and DMSO); they also analyzed tricine osmotic properties and its effect on both ice crystal formation and oxidative damage occurring in cryopreservation. No direct measurements of ROS production was carried out, however the levels of SOD, catalase and a general antioxidant capacity (by DPPH method) was measured in the presence of tricine. In addition, some properties (ESR and ATPase activitiy) of thawed RBCs were also investigated. Finally, they attempted to compare different CPAs by means of mathematical models. Despite, I found this latter part of less interest, my general opinion is that the study appears well planned and reports interesting results. However, some specific (minor?) points could be improved in the manuscript. Please find below details.
Please specify the abbreviation IRI, DSC, MLGC... not only in the figure caption, but also the first time they appear in the text.
In fig. 4, n-value should be provided, as well as the indication of SE or SD.
I suggest to change the 2.4 title
Some sentences could/should be rephrased, see for example the ones in lines 83-87, 120-121, 191-192, 277-278.
What exactly is the control group in fig. 5? And why this is absent in 5A and 5D? Again, what about PBS in Fig. 6C and D?
In my opinion, the scheme reported in Fig. 3D did not add anything to the manuscript; moreover, the same information is also summarized in Fig. 7.
M&M section:
- Except for 4.7, 4.9 and 4.11 sections, no references were reported for the methods applied. If already used, citing the previous paper(s) avoid a more detailed description of procedures (otherwise these should be reported).
- please convert “rpm” in “x g”.
In some cases, Fig. 2 and Fig. 4, other statistical analysis as ANOVA could have allowed to compare also the effect(s) among the different tricine concentrations. This could also be of interest.
Minors:
Line 401: it is
Line 127: I suggest to use the word “representative” for images reported in Figs 3,4, and 6.
Line 220: insert space after [45]
Author Response
Please see the file "Response to Reviewer 2"

Reviewer 3 Report
Dear Authors,
The authors report the use of a new cryoprotectant, tricine, to cryopreserve red blood cells (RBC). Tricine increased the RBC recovery my 4 times, compared to traditional cryoprotectant (dimethyl sulfoxide, hydroxyethyl starch). Tricine reduced the size of the crystal and the crystal formation. In addition, Tricine decrease the oxidative damage and maintain the enzymatic function of the antioxidant enzymes, present in the RBC. The manuscript is well written, and the subject is very important in the field of blood transfusion.
I have some comments:
Major:
- The authors must provide the protocol number and which IACUC committee approved the animal study.
- How many RBC were cryopreserved and maintained in the fridge?
- An important point raised by the authors is the capacity to build a bank of RBC and thaw them in case of needs. However, in the materials and methods, it is not clear for how long the RBC were stored in the nitrogen liquid tank. This must be mentioned.
- What is the viability of the RBC after isolation, and the enzymes activity, reported in figure 5 and 6?
- Freshly isolated RBC must be a reference for all the data provided in the figure 3, 5,6
- Data about the RBC morphology in figure 3 and figure 6 A/B seem to be similar. In fact, rather than to try to mimic approximately the solution tonicity by having PBS at 2.9% of NaCl, the authors should present the morphology of RBC like in figure 6A/B, by including the different groups, DMSO and HES. It will be more representative of the RBC morphology.
- The maintain of the morphology and the enzymes activities are very important for RBC function. However, one of the main functions of the RBC is to carry Oxygen and Carbon Dioxide. If RBC can’t carry O2/CO2, RBC can’t be transfused. The authors must show that the RBC can still carry O2/CO2.
- How quickly after thawing, the RBC were characterized and analyzed? If it is just after thawing, often long-term cell damage cannot be detected. It is important to see if the RBC can maintain their function after 1-3 days after thawing.
- In table S2, S3 and S4, why when we look at the values, reading horizontally we have a certain value and a different one when we read vertically (e.g: S4, HES to DMSO is 1/3 horizontally, and HES to DMSO is 1 vertically)? Should not the values the same?
Minor:
- The color of each group in figure 1 should be the same on A and B, to facilitate the reading. Same comment for another figure like figure 5.
- ESR should be spell out the first time it is used.
- In table 1 title, the authors mention 5 criteria, but I can see only 4. Could the authors double check and correct the error?
- Table S7: Where is the reference about the Trehalose?
Author Response
Please see the file "Response to Reviewer 3"

Round 2
Reviewer 1 Report
"The authors are to be commended for the significant amount of additional work they have undertaken in response to my comments on the original manuscript. All the points raised in my original review have been more than satisfactorily answered in their response and the changes made by them to their manuscript. There is one minor typographical error on line 55: “Glycerol is the only CPA that approved for clinical cryopreservation...”"Reviewer 3 Report
Dear authors,
I have no additional comments.
Thanks.